# DG–CA3 circuitry mediates hippocampal representations of latent information

Alexandra T. Keinath [1✉], Andrés Nieto-Posadas[1], Jennifer C. Robinson[2] & Mark P. Brandon [1,3✉]

Survival in complex environments necessitates a flexible navigation system that incorporates memory of recent behavior and associations. Yet, how the hippocampal spatial circuit represents latent information independent of sensory inputs and future goals has not been determined. To address this, we image the activity of large ensembles in subregion CA1 via wide-field fluorescent microscopy during a novel behavioral paradigm. Our results demonstrate that latent information is represented through reliable firing rate changes during unconstrained navigation. We then hypothesize that the representation of latent information in CA1 is mediated by pattern separation/completion processes instantiated upstream within the dentate gyrus (DG) and CA3 subregions. Indeed, CA3 ensemble recordings reveal an analogous code for latent information. Moreover, selective chemogenetic inactivation of DG–CA3 circuitry completely and reversibly abolishes the CA1 representation of latent information. These results reveal a causal and specific role of DG–CA3 circuitry in the maintenance of latent information within the hippocampus.

[1] Department of Psychiatry, Douglas Hospital Research Centre, McGill University, 6875 Boulevard LaSalle, Verdun, QC H4H 1R3, Canada. [2] Department of Psychological and Brain Sciences, Rajen Kilachand Center for Integrated Life Sciences and Engineering, Boston University, 610 Commonwealth Avenue, Boston, MA 02215, USA. [3] Integrated Program in Neuroscience, Montreal Neurological Institute, McGill University, 3801 University Street, Montreal, QC H3A 2B4, Canada. ✉email: atkeinath@gmail.com; mark.brandon@mcgill.ca

Hippocampal subregions CA1 and CA3 represent space through the coordinated activity of place cells—sparsely active cells tuned to different preferred locations that tile the navigable space, collectively forming a hippocampal map specific to each context[1]. Changes to sensory cues or cognitive demands within a context can lead to widespread changes in the firing rates of place cells[2–8], a type of hippocampal pattern separation known as rate remapping[5]. Previous work has demonstrated that different sources of information and circuit mechanisms are capable of driving rate remapping: sensory-driven remapping[5] is mediated in part by both lateral entorhinal[9] and trisynaptic[10] circuits, while goal-oriented remapping[4,7,8] is mediated by a prefrontal-thalamic circuit[6] which is distinct from the lateral entorhinal and trisynaptic circuits.

However, a flexible navigational system should encode aspects of the current context beyond immediate sensory input and future goals. Indeed, latent information, such as a memory of recent behavior or experiences independent of future goals, can be especially important for discovering and representing relationships that extend beyond the capacity of immediate sensory information. Yet, whether and how the hippocampus represents latent information, as well as the neural circuitry that maintains these representations in the absence of continuous sensory information or goal-directed behavior, has not been determined.

Here we show that hippocampal subregions CA1 and CA3 represent latent information through changes in firing rate during unconstrained 2D navigation in a novel behavioral paradigm in the absence of explicit task demands. Next, we demonstrate that chemogenetic inactivation of DG–CA3 reversibly abolishes this representation. Together, these results demonstrate a causal and specific role of DG–CA3 circuitry in the representation of latent information during unconstrained navigation.

## Results

**Paradigm and neural recording**. We first developed a simple behavioral paradigm to characterize whether and how latent information is represented during unconstrained 2D navigation. To this end, we designed a single-compartment arena accessible via two entryways connected by a short hallway (Fig. 1a). Previous results have demonstrated that point of entry can modulate hippocampal representations across different compartments[11–13]. In our paradigm, entry to the single compartment through either doorway is associated with a distinct recent history including the animal's spatial location, direction of running within the hallway, and the turns made as the mouse enters the larger 2D compartment. Once inside the compartment, however, the spatial position and sensory cues available to the animal are identical regardless of the prior point of entry; thus any remapping within the room can be attributed to latent information.

We imaged the calcium dynamics of large populations of CA1 neurons using a miniaturized head-mounted wide-field microscope (miniscope.org) as mice freely explored this environment for 20 min[14,15] (Fig. 1b; $n = 6$ mice, one mouse excluded for persistent entryway behavioral bias, $n = 5$ analyzed, 29 sessions; Supplementary Fig. 1, Supplementary Table 1). Following motion correction[16], cells were segmented and calcium traces were extracted via constrained nonnegative matrix factorization[17,18] (Supplementary Figs. 2, 3). The likelihood of spiking events, a correlate of firing rate[19], was inferred from deconvolution of the filtered calcium traces via a second-order autoregressive model (Fig. 1c). All further analyses were conducted on the inferred likelihood of spiking events. Next, we extracted times when the mouse was in the compartment and partitioned these data according to both the most recent entryway and session half. From these data, we identified place cells as cells which demonstrated reliable spatial correlations across session halves either after entering from the same entryway or different entryways (correlations greater than the 95th percentile relative to a shuffled control, corrected for multiple comparisons, see "Methods" section; Fig. 1d, Supplementary Figs. 4, 5). Importantly, because biases in spatial sampling can be correlated with the most recent entryway, we subsampled the data to match the

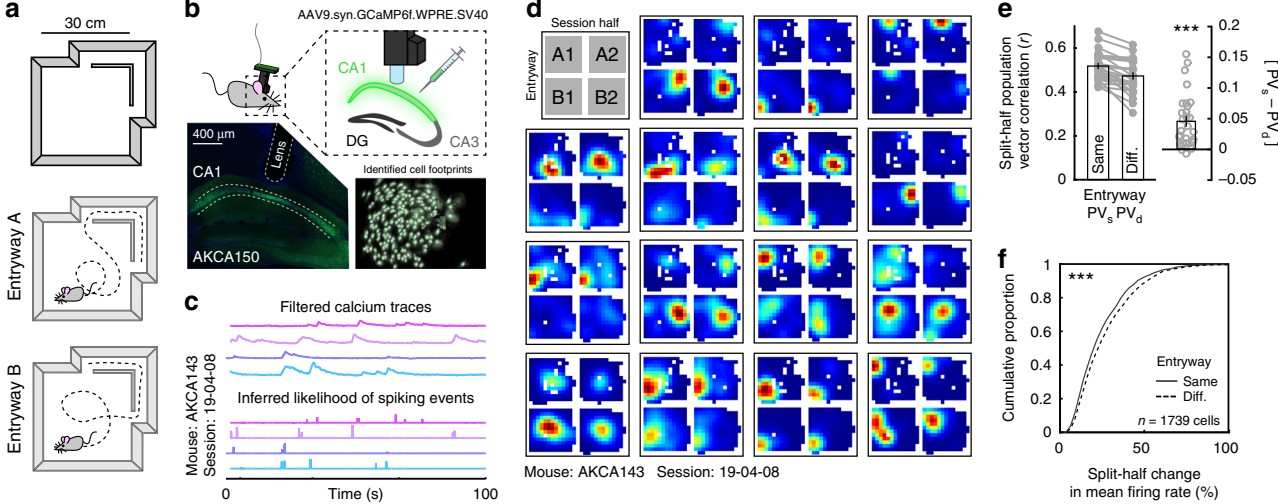

**Fig. 1 The CA1 rate code depends on entryway. a** Schematic of the recording environment. **b** One-photon calcium recording setup, with an example of viral expression (lower left) and the spatial footprints of cells extracted from the in vivo recorded calcium signal for an example session (lower right). **c** Example of extracted filtered calcium traces (top) and the resulting inferred likelihood of spiking events (bottom). **d** Example rate maps from 15 simultaneously recorded place cells when the data are divided by most recent entryway and session half. Rate maps are normalized from zero (blue) to the peak (red) across all four maps. **e** Split-half correlations of population activity within the compartment when the mouse entered from the same versus the different entryway. Each dot represents one session from one mouse. Data are from 29 sessions from 5 mice. Correlations were significantly higher when the mouse entered from the same entryway (Wilcoxon signed-rank test: $Z = 4.573$, $p = 4.80e-6$). **f** Cumulative distribution of split-half changes of mean firing rates within the compartment when the mouse entered from the same versus the different entryway. Mean firing rates were significantly more similar when the mouse entered from the same entryway (Wilcoxon rank-sum test: $Z = 4.209$, $p = 2.56e-5$). All bar graphs reflect mean ± 1 SEM; $p$-values are uncorrected and two-sided. Data from (**e**, **f**) provided as Source data file. ***$p < 0.001$.

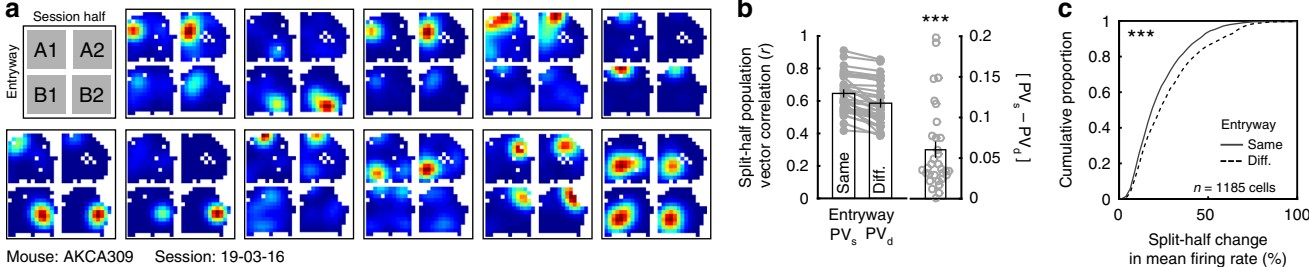

**Fig. 2 The CA3 rate code depends on entryway. a** Example rate maps from 11 simultaneously recorded place cells when the data are divided by entryway and session half. Rate maps are normalized from zero (blue) to the peak (red) across all four maps. **b** Split-half correlations of population activity within the compartment when the mouse entered from the same versus the different entryway. Each dot represents one session from one mouse. Data are from 32 sessions from 4 mice. Correlations were significantly higher when the mouse entered from the same entryway (Wilcoxon signed-rank test: $Z = 4.937$, $p = 7.95e-7$). **c** Cumulative distribution of split-half changes of mean firing rates within the compartment when the mouse entered from the same versus the different entryway. Mean firing rates were significantly more similar when the mouse entered from the same entryway (Wilcoxon rank-sum test: $Z = 5.150$, $p = 2.61e-7$). All bar graphs reflect mean ± 1 SEM; $p$-values are uncorrected and two-sided. Data from (**b**, **c**) provided as Source data file. ***$p < 0.001$.

spatial sampling distributions across entryways and halves prior to all analyses (Supplementary Figs. 6, 7; see "Methods" section).

**The CA1 rate code represents most recent entryway.** If point of entry drives remapping within an environment, then maps of the compartment should be more similar across session halves when the mouse entered from the same entryway as opposed to different entryways. To quantify representational similarity, we computed the mean split-half correlation of the place cell population vector following entry through either the same entryways ($PV_s$) or different entryways ($PV_d$). This analysis revealed that, at a population level, the CA1 compartment map was more similar when the mouse entered through the same entryway than when he entered through different entryways (Fig. 1e, Supplementary Fig. 8). Further analysis revealed that these population-level differences reflected reliable differences in place cell firing rates (Fig. 1f, Supplementary Fig. 9) in the absence of substantial changes in preferred locations (Supplementary Fig. 10), consistent with rate remapping[5]. The orientation of the CA1 map was not modulated by entryway (Supplementary Fig. 10). Remapping was robust when selecting place cells via other criteria (Supplementary Fig. 11), most pronounced in the place cell population (Supplementary Fig. 12), and robust to other quantifications of calcium transients (Supplementary Fig. 13). The magnitude of entryway remapping was not predicted by behavioral or signal-to-noise (SNR) covariates (Supplementary Fig. 14). Entryway remapping effects were also observed without matching sampling distributions prior to computing comparisons (Supplementary Fig. 15). Similar results were observed in a larger environment in a separate cohort of mice (Supplementary Fig. 16). Together these results demonstrate coding of recent history through remapping of the CA1 rate code during unconstrained 2D navigation. For completeness, the CA1 map of the hallway also remapped according to point of entry (Supplementary Fig. 17), as expected given the typical observation of direction-selective coding in linear track environments[20].

**The CA3 rate code represents most recent entryway.** Previous work has demonstrated that the trisynaptic circuit partially mediates remapping of the CA1 rate code when immediate sensory cues are available[10]. We hypothesized that CA1 coding of latent variables may also be driven by trisynaptic input. If so, similar remapping should also be observed in upstream CA3. To test this possibility, we repeated our experiment and analyses while recording place cells from right CA3 ($n = 4$ mice, 32 sessions; Fig. 2a, Supplementary Figs. 1–6; Supplementary Table 1).

Computing $[PV_s - PV_d]$ revealed that the CA3 compartment map was more similar when the mouse entered through the same entryway than when he entered through a different entryway (Fig. 2b, Supplementary Fig. 8). Again, this difference reflected reliable differences in place cell firing rates (Fig. 2c, Supplementary Fig. 9) without substantial changes in preferred locations (Supplementary Fig. 10). The orientation of the CA3 map was not modulated by entryway (Supplementary Fig. 10). Remapping was robust when selecting place cells via other criteria (Supplementary Fig. 11), most pronounced in the place cell population (Supplementary Fig. 12), and robust to other quantifications of calcium transients (Supplementary Fig. 13). The magnitude of entryway remapping was not predicted by behavioral or signal-to-noise (SNR) covariates (Supplementary Fig. 14). Entryway remapping effects were also observed without matching sampling distributions prior to computing comparisons (Supplementary Fig. 15). Together these results indicate reliable remapping of the CA3 rate code on the basis of the most recent point of entry, similar to that of the CA1 place code. For completeness, the CA3 map of the hallway also remapped according to point of entry (Supplementary Fig. 17), as expected given the typical observation of direction-selective coding in linear track environments[20].

**Coding is entry-specific and persists across time and space.** Other latent variables, such as distant prior experience or planned future behavior, may also be represented via remapping of the CA1 and/or CA3 rate codes. Indeed, in goal-oriented tasks both retrospective and prospective rate coding have been observed throughout the broader hippocampal formation[7,8]. We tested this possibility in our data by repeating our analysis when segmenting the behavioral trajectory according to the second-most recent entryway into the compartment (Two-back) and the exitway (Fig. 3a). In both CA1 and CA3, only segmentation by the most recent entryway yielded reliable representation. Together this suggests that recent prior experience, but not distant experience or planned future behavior, is coded during navigation in the absence of task demands.

If the representational differences we observe reflect the sustained influence of latent entryway information, then these differences should persist across locations within the compartment and across behaviorally-relevant timescales. To test these possibilities, we first computed entryway $[PV_s - PV_d]$ separately at each location in the compartment (Fig. 3b). Reliable differences in map similarity were observed across the compartment in both the CA1 and CA3. The magnitude of remapping was negatively correlated with the distance to the nearest entryway in CA1 (Pearson's correlation, $r = -0.279$, $p = 4.920e-4$) but not CA3

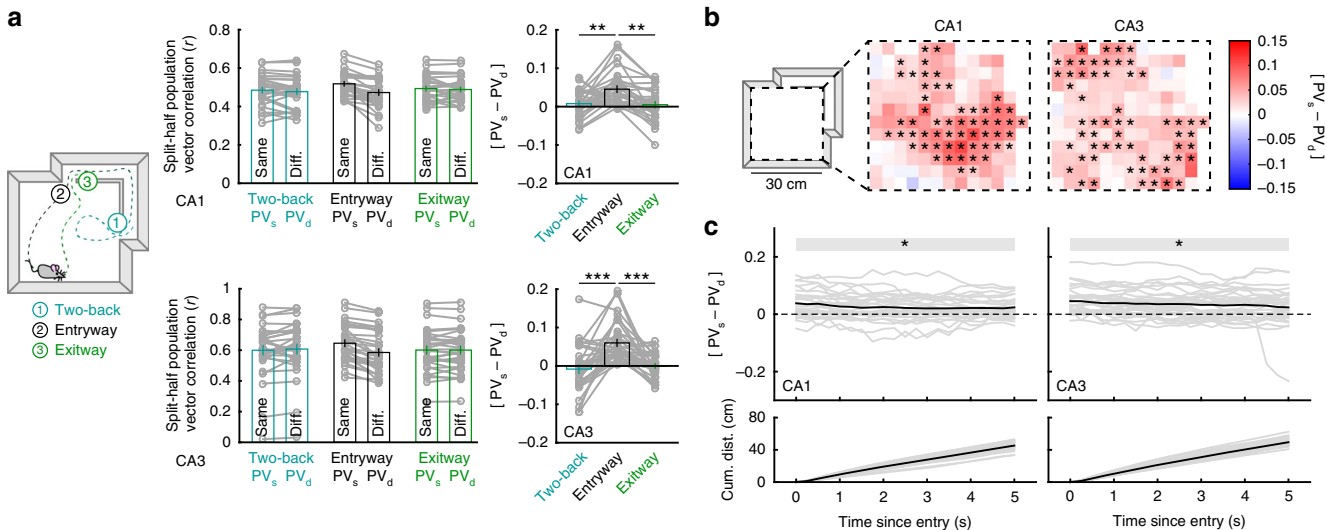

**Fig. 3 Remapping is specific to entryway and persists across space and time. a** Schematic of trajectory segmentation by the second-most recent entryway into the compartment (Two-back), the most recent entryway, and exitway (left). Split-half correlations of population activity within the compartment when the data are partitioned according to various segmentations for initial CA1 (top) and CA3 (bottom) recordings. Only partitioning by entryway yielded reliable remapping (CA1: Wilcoxon signed-rank test versus 0, Two-back: $Z = 1.4812$, $p = 0.139$; Exitway: $Z = 0.7082$, $p = 0.4822$; Wilcoxon rank-sum test between conditions, Two-back versus Entryway: $Z = 2.6437$, $p = 0.0082$; Entryway versus Exitway: $Z = 3.2658$, $p = 0.0011$; Two-back versus Exitway: $Z = 0.4510$, $p = 0.652$; CA3: Wilcoxon signed-rank test versus 0, Two-back: $Z = 0.9349$, $p = 0.350$; Exitway: $Z = 0.0187$, $p = 0.985$; Wilcoxon rank-sum test between conditions, Two-back versus Entryway: $Z = 4.5182$, $p = 6.24e-6$; Entryway versus Exitway: $Z = 5.0822$, $p = 3.73e-7$; Two-back versus Exitway: $Z = 1.0003$, $p = 0.317$). **b** Median $[PV_s - PV_d]$ computed separately at each location within the compartment for data recorded in CA1 and CA3. Significance markers indicated outcomes of uncorrected Wilcoxon signed-rank tests. **c** $[PV_s - PV_d]$ when only including data from progressively later times since entering the compartment for data recorded in CA1 and CA3 (top). For example, values at 5 s were computed using only data recorded at least 5 s after the mouse entered the compartment. Computed at every 0.25 s increment. Shaded regions containing significance markers indicate all increments for which the outcome of an uncorrected Wilcoxon signed-rank test versus 0 is p < 0.05. Mean cumulative distance traveled since entering the compartment (bottom). Each gray line denotes a single session. All bar graphs reflect mean ± 1 SEM; p-values are uncorrected and two-sided. Data provided as Source data file. *p < 0.05, **p < 0.01, ***p < 0.001.

($r = 0.009$, $p = 0.917$). The magnitude of remapping was not correlated with differences in median speed across locations (Pearson's correlation, CA1: $r = 0.127$, $p = 0.139$; CA3: $r = 0.046$, $p = 0.595$). Next, we again computed $[PV_s - PV_d]$ over the entire compartment, while including data only from progressively later times since the mouse entered the compartment. This analysis revealed that reliable remapping was observed when including only data recorded at least 5 s since entering the compartment in both CA1 and CA3 (Fig. 3c, Supplementary Fig. 6), well beyond the timescales of both neural activity and calcium reporter dynamics. Together these results indicate that the rate remapping we observed in CA1 and CA3 reflects a representation of the most recent entryway which persists across both space and time.

**Inactivation of DG–CA3 abolishes entry coding in CA1.** If CA1 rate remapping in this paradigm is driven by trisynaptic input, then disruption of the trisynaptic circuit should eliminate such remapping. If, on the other hand, remapping of the CA1 and CA3 place codes is driven by common input originating outside of the hippocampus, then inhibition of the trisynaptic circuit should spare this remapping. To causally adjudicate between these possibilities, we repeated our experiment while recording from right CA1 and simultaneously manipulating the trisynaptic circuit in a new cohort of mice. To this end, we employed a Grik4-cre mouse line which expresses cre in both the dentate gyrus and CA3, but not CA1[21]. These regions were transfected bilaterally with the cre-mediated inhibitory designer receptor hm4di[22] ($n = 3$ mice; Fig. 4a, Supplementary Fig. 18). On interleaved trials, either clozapine-N-oxide (CNO), the designer drug which activates hm4di, or saline, the vehicle control, was administered intraperitoneally one hour and fifteen minutes prior to recording (19

Saline sessions, 19 CNO sessions, Supplementary Figs. 1–4; Supplementary Table 1). An additional control cohort not expressing hm4di was also recorded with the same paradigm ($n = 3$ mice, 14 CNO sessions, 13 saline sessions, Supplementary Figs. 1–4; Supplementary Table 1).

Inhibition of the trisynaptic circuit eliminated remapping of the CA1 place code by entryway as assayed by multiple measures (Fig. 4b, Supplementary Fig. 19). Computing $[PV_s - PV_d]$ revealed that the CA1 compartment map was more similar when the mouse entered through the same entryway than when he entered through a different entryway in all conditions except for the CNO with hm4di condition (Fig. 4c, Supplementary Fig. 8). Furthermore, $[PV_s - PV_d]$ of the CNO with hm4di condition was reliably decreased relative to all other conditions and was not significantly larger than zero. Importantly, $PV_s$ remained equally high in all conditions (Wilcoxon rank-sum tests: Zs < 1.0920, ps > 0.274), suggesting that the lack of remapping during CNO with hm4di sessions was not driven by a disruption of the CA1 place code more generally (Fig. 4c). Consistent with this, other coding properties such as mean and peak firing rates, spatial information content, and overall split-half reliability did not reliably differ during Saline versus CNO sessions for with hm4di mice (Supplementary Fig. 20). At an individual cell level, mean firing rates were reliably more similar when entering through same entryway than different entryways in all conditions except for the CNO with hm4di condition (Fig. 4d, Supplementary Fig. 9). These effects were qualitatively robust when selecting place cells via other criteria (Supplementary Fig. 11), most pronounced in the place cell population (Supplementary Fig. 12), and robust to other quantifications of calcium transients (Supplementary Fig. 13). The magnitude of these effects was not predicted by behavioral or signal-to-noise (SNR) covariates (Supplementary

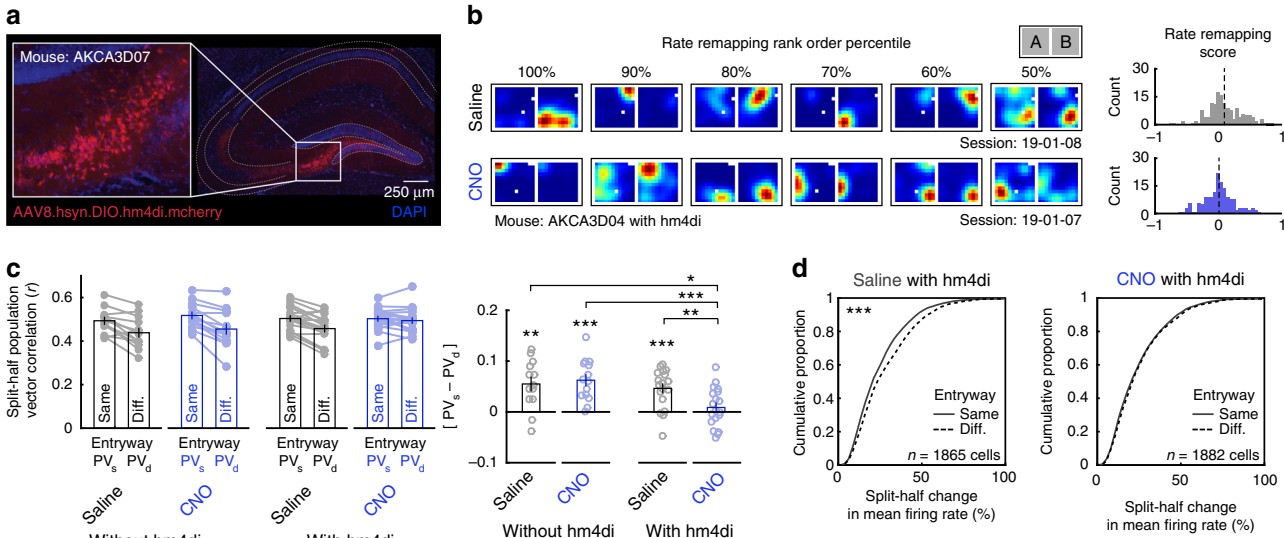

**Fig. 4 Inhibition of DG–CA3 circuitry eliminates remapping of the CA1 rate code. a** Representative example of cre-mediated hm4di-mcherry transfection in left dorsal DG and CA3 (see Supplementary Fig. 18 for additional examples and quantification details). **b** Six example rate maps of simultaneously recorded place cells from consecutive Saline/CNO sessions selected and ordered by their rate remapping score rank (left). Rate maps are normalized from zero (blue) to the peak (red) across both maps. Distribution of rate remapping scores and mean score for these example sessions (right). Rate remapping score defined as split-half change in mean firing rate for different minus same entryway. **c** Split-half correlations of population activity within the compartment when the mouse entered from the same versus the different entryway for all conditions. Correlations were significantly higher when the mouse entered from the same entryway in all conditions except for CNO with hm4di (Wilcoxon signed-rank test: without hm4di Saline: $p = 0.0046$; without hm4di CNO: $p = 1.2e{-}4$; with hm4di Saline: $Z = 3.380$; $p = 7.2e{-}4$; with hm4di CNO: $Z = 0.9256$, $p = 0.355$). Correlation differences were significantly lower in the CNO with hm4di condition than all other conditions (Wilcoxon rank-sum tests: without hm4di Saline versus without hm4di CNO: $Z = 0.267$, $p = 0.790$; without hm4di Saline versus with hm4di Saline: $Z = 0.691$, $p = 0.490$; without hm4di CNO versus with hm4di Saline: $Z = 1.147$, $p = 0.251$; without hm4di Saline versus with hm4di CNO: $Z = 2.379$, $p = 0.017$; without hm4di CNO versus with hm4di CNO: $Z = 3.479$, $p = 5.0e{-}4$; with hm4di Saline versus with hm4di CNO: $Z = 2.861$, $p = 4.2e{-}3$). **d** Cumulative distribution of split-half changes of mean firing rates within the compartment when the mouse entered from the same versus the different entryway for mice with hm4di. Mean firing rates were significantly more similar when the mouse entered from the same entryway during Saline sessions (Wilcoxon signed-rank test: $Z = 5.052$, $p = 4.4e{-}7$), but not during CNO sessions ($Z = 1.024$, $p = 0.306$). All bar graphs reflect mean $\pm 1$ SEM; $p$-values are uncorrected and two-sided. Data from (**c**, **d**) provided as Source data file. $*p < 0.05$, $**p < 0.01$, $***p < 0.001$.

Fig. 14), and similar results were observed without matching sampling distributions prior to computing comparisons (Supplementary Fig. 15). Moreover, the elimination of entryway remapping in mice with hm4di under CNO persisted across time and sessions (Supplementary Fig. 21). Remapping of the hallway place code persisted across all conditions (Supplementary Fig. 22).

## Discussion

Characterizing the neural basis of navigation has been a major focus of research since the discovery of hippocampal place cells; yet a mechanistic explanation of how hippocampal spatial circuits represent information beyond sensory inputs and goal-oriented behavior has been limited, likely hindered by difficulties assaying latent information isolated from a behavioral task. Here we used a novel behavioral paradigm aimed to examine how the hippocampus encodes latent information in a 2D environment. We first report that latent information, in the form of the most recent entryway in a multientry compartment, is robustly encoded by rate changes in the CA1 place cell representation in the absence of explicit task demands. Building on proposals that CA3 recurrent connections are theoretically capable of maintaining information in the absence of continuous input via attractor dynamics, and that the DG is particularly well-suited to disambiguate similar input representations via pattern separation, we hypothesized the DG–CA3 circuit mediated the representation of latent information in this paradigm[23–25]. Recordings in CA3 revealed that this circuit also maintained an analogous code for latent information,

indicating that such a role was plausible. Finally, to causally test this hypothesis, we used a transgenic mouse line and chemogenetics to reversibly inactivate the DG–CA3 circuit while recording from region CA1. These experiments revealed that the representation of latent information in CA1 is completely dependent on CA3–DG activity in this paradigm. Importantly, the spatial tuning of the CA1 representation remained intact during DG–CA3 inactivation, offering support to the hypothesis that the key role of the trisynaptic circuit (EC–DG–CA3–CA1) may not be spatial coding per se (similar to the results of ref. [26]), but rather the encoding of additional information beyond a navigator's immediate spatial location.

Our observation of entryway coding in the absence of task demands prompts further questions about the specific source and ethological relevance of such information. One possibility is that the remapping we observe in our paradigm may be driven by the stereotyped behavior of entering from the hallway, where navigation is largely constrained. Prior work has demonstrated that direction of entry is a powerful disambiguating cue capable of modulating the similarity of hippocampal maps across repeated connected compartments[11–13]. Moreover, we observed strong remapping within the hallway, and the strength of this remapping was generally correlated with the extent of entryway remapping in the compartment across experiments. Thus, each entry was preceded by a unique neural state space. This suggests that entryway remapping may follow from reinforcement learning models, where transitions in hippocampal state space are governed by repeated experience with the navigational structure of the

environment[27], and that DG–CA3 circuitry may be critical to maintaining this information in the absence of immediate sensory cues. At an ethological level, entryway remapping, and representation of latent variables during unconstrained navigation more generally, may be crucial for flexibly discovering and representing contingencies during navigation. For example, one could imagine that trajectories originating from one burrow through an open space may be associated with potential predators, while entry from an alternate borrow may subsume less risk. Finally, we note that repeated experience may also influence the degree of entryway remapping. In our data, entryway remapping persevered for up to 21 sessions of recording and following repeated exposure to a fixed map in the case of mice with hm4di under CNO. Nevertheless, experience on longer timescales may serve to stabilize the compartment map.

At a theoretical level, our results contribute to a growing literature demonstrating that hippocampal rate remapping can be driven by multiple sources, potentially mediated by distinct circuit mechanisms. Previous studies have demonstrated that rate remapping can be driven by sensory inputs[5] and that such representation partially depends on both the lateral entorhinal cortex[9] and NDMA receptors in the DG[10], though on average these studies report a 20% and 50% reduction in rate remapping, respectively. Importantly, such paradigms are not designed to rule out a latent/mnemonic contribution to rate remapping; instead, these paradigms may plausibly assay contributions of both sensory inputs and latent information. In the current study in which we rule out the possibility of sensory-driven remapping, inactivation of DG–CA3 led to an average 86% reduction in remapping, with residual remapping failing to significantly differ from chance. Moreover, in our data remapping within the hallway persisted even in experimental mice with hm4di under CNO, suggesting that hallway remapping and remapping by entryway within the compartment are driven by dissociable mechanisms. In the hallway, movement is constrained such that external cues are consistently experienced in a set pattern. Thus within the hallway, remapping may be driven by repeated experience with immediately-available sensory cues, as is typically seen on linear tracks[28]. Once within the compartment, however, behavior is unconstrained and immediate external cues therefore do not disambiguate the most recent entryway. Taken together, it is possible that the lateral entorhinal cortex and the DG–CA3 circuit mediate distinct sensory- and latent- contributions to rate remapping.

In a similar vein, previous studies demonstrating rate remapping on the basis of goal-oriented behavior have not disambiguated a contribution of latent information from a contribution of planned behavior[4,7,8], and have demonstrated that remapping reflecting planned behavior is primarily driven by the nucleus reuniens, which bypasses the DG–CA3 circuit[6]. Consistent with this distinction we observed no coding of future exitway, a component which has been observed in goal-oriented tasks[6–8], suggesting that the DG–CA3-dependent remapping we observed in this paradigm did not reflect planned behavior. We anticipate that a similar experimental approach will help to address several lingering questions, including the relationship between latent, sensory, and goal-oriented rate remapping, the specificity of trisynaptic, lateral entorhinal, and prefrontal-thalamic circuit contributions to these forms of rate remapping, and the contribution of the trisynaptic circuit to other forms of hippocampal remapping[5,29,30].

## Methods

**Subjects.** Naive male mice (Grik4-cre transgenic mouse line C57BL/6-Tg(Grik4-cre)G32-4Stl/J, The Jackson Laboratory; all other mice C57Bl/6, Charles River) were housed individually on a 12-h light/dark cycle at 22 °C and 40% humidity

with food and water ad libitum. All experiments were carried out during the light portion of the light/dark cycle, and in accordance with McGill University and Douglas Hospital Research Centre Animal Use and Care Committee (protocol #20157725) and in accordance with Canadian Institutes of Health Research guidelines. One mouse was excluded from the initial CA1 recording experiments prior to analysis due to an extreme entryway bias (>5:1) which persisted across all sessions.

**Surgeries.** During all surgeries, mice were anesthetized via inhalation of a combination of oxygen and 5% Isoflurane before being transferred to the stereotaxic frame (David Kopf Instruments), where anesthesia was maintained via inhalation of oxygen and 0.5–2.5% Isoflurane for the duration of the surgery. Body temperature was maintained with a heating pad and eyes were hydrated with gel (Optixcare). Carprofen (10 ml kg$^{-1}$) and saline (0.5 ml) were administered subcutaneously at the beginning of each surgery. Preparation for recordings involved three surgeries per mouse.

First, at the age of six to ten weeks, each mouse was transfected with a 400 nl injection of the calcium reporter GCaMP6f according to the specific viral construct and injection coordinates described in Supplementary Table 2. The original titre of the AAV9.syn.GCaMP6f.WPRE.SV40 construct, sourced from University of Pennsylvania Vector Core, was 3.26e14 GC-ml and was diluted in sterile PBS (1 part virus to 1 parts PBS) before surgical microinjection. The original titre of the AAV5.CaMKII.GCaMP6f.WPRE.SV40 construct, sourced from Addgene, was 2.3e13 GC-ml and was diluted in sterile PBS (1 part virus to 2 parts PBS) before surgical microinjection. For chemogenetic experiments, mice expressing Grik4-Cre were also injected bilaterally with the cre-mediated Designer Receptor Exclusively Activated by Designer Drugs (DREADDs) viral construct AAV8.hsyn.DIO.hm4di.mcherry according to the injection schedule described in Supplementary Table 3 during this surgery. The original titre of this virus, sourced from Addgene, was 4.3e12 GC-ml and was diluted in sterile PBS (5 parts virus to 6 parts PBS) before surgical microinjection. All injections were administered via glass pipettes connected to a Nanoject II (Drummond Scientific) injector at a flow rate of 23 nl s$^{-1}$.

One week post-injection, either a 1.8 mm or 0.5 mm diameter gradient refractive index (GRIN) lens (Go!Foton) was implanted above either dorsal CA1 or CA3 as indicated in Supplementary Table 2. Implantation of the 1.8 mm diameter GRIN lens required aspiration of intervening cortical tissue, while no aspiration was required for implantation of the 0.5 mm diameter GRIN lens. Results observed using 1.8- or 0.5-mm diameter grin lenses were similar. In addition to the GRIN lens, two stainless steel screws were threaded into the skull above the contralateral hippocampus and prefrontal cortex to stabilize the implant. Dental cement (C&B Metabond) was applied to secure the GRIN lens and anchor screws to the skull. A silicone adhesive (Kwik-Sil, World Precision Instruments) was applied to protect the top surface of the GRIN lens until the next surgery.

One to three weeks after lens implantation, an aluminum baseplate was affixed via dental cement (C&B Metabond) to the skull of the mouse, which would later secure the miniaturized fluorescent endoscope (miniscope) in place during recording. The miniscope/baseplate was mounted to a stereotaxic arm for lowering above the implanted GRIN lens until the field of view contained visible cell segments and dental cement was applied to affix the baseplate to the skull. A polyoxymethylene cap with a metal nut weighing ~3 g was affixed to the baseplate when the mice were not being recorded, to protect the baseplate and lens, as well as to simulate the weight of the miniscope.

After surgery, animals were continuously monitored until they recovered. For the initial three days after surgery mice were provided with a soft diet supplemented with Carprofen for pain management (MediGel CPF, ~5 mg kg$^{-1}$ each day). Screening and habituation to recording in a rectangular 20 × 40 cm semi-transparent plastic home cage environment began 3–7 days following the baseplate surgery and continued for at least 2 days until the quality and reliability of the calcium data were confirmed.

**Data acquisition.** In vivo calcium videos were recorded with a miniscope (v3; miniscope.org) containing a monochrome CMOS imaging sensor (MT9V032C12STM, ON Semiconductor) connected to a custom data acquisition (DAQ) box (miniscope.org) with a lightweight, flexible coaxial cable[14]. The DAQ was connected to a PC with a USB 3.0 SuperSpeed cable and controlled with Miniscope custom acquisition software (miniscope.org). The outgoing excitation LED was set to between 3–6%, depending on the mouse to maximize signal quality with the minimum possible excitation light to mitigate the risk of photobleaching. Gain was adjusted to match the dynamic range of the recorded video to the fluctuations of the calcium signal for each recording to avoid saturation. Behavioral video data were recorded by a webcam mounted above the environment. Behavioral video recording parameters were adjusted such that only the red LED on the CMOS of the miniscope was visible. The DAQ simultaneously acquired behavioral and cellular imaging streams at 30 Hz as uncompressed avi files and all recorded frames were timestamped for post-hoc alignment.

All recording environments were constructed of a gray Lego base and black Lego bricks (Lego, Inc) according to the dimensions specified in the main text and Supplementary Figures. All external walls had a height of 22 cm; all internal walls

had a height of 15 cm. All hallways were 5 cm wide; due to the width mice typically ran the length of the hallway rather than turning around in the hallway. During recording, the environment was dimly lit by a nearby computer screen, which served as directional cue. A white-noise generator was placed above the environment to mask uncontrolled ambient sounds. Each recording session lasted 20 min, and only one session was recorded per day to avoid photobleaching. The mouse was always placed in the corner of the hallway at the start of the session and was allowed to explore the environment for 15–30 s prior to data acquisition. Following each recording the environment was cleaned with disinfectant (Prevail).

For CA3 inactivation experiments, 5 mg kg$^{-1}$ of clozapine-N-oxide (CNO + 0.7% DMSO) was injected Intraperitoneally 1 h and 15 min prior to recording. Mice were returned to their home cage between the injection and the start of the recording session. We conducted two separate control experiments to rule out the possibilities that our results could be explained by injection procedure, expression of hm4di-mcherry, or to non-specific effects of CNO itself. As our first within-animal control, we injected sterile saline instead of CNO and repeated the recording and analysis procedures (Fig. 4b–d). The order of Saline and CNO recordings was interleaved within mouse, and whether the first recording session for a mouse followed a Saline or CNO injection was randomized. As a second control, to ensure that any differences between Saline and CNO sessions were attributable to the interaction between hm4di and CNO, and not an effect of CNO or its metabolites alone, CNO injection experiments were repeated in a second across-mouse control group which did not express hm4di (Fig. 4c).

**Data preprocessing**. Calcium imaging data were preprocessed prior to analyses via a pipeline of open source MATLAB (MathWorks; version R2015a) functions to correct for motion artifacts[16], segment cells and extract transients[17,18], and infer the likelihood of spiking events via deconvolution of the transient trace through a second-order autoregressive model[19]. The motion-corrected calcium imaging data were manually inspected to ensure that motion correction was effective and did not introduce additional artifacts. Following this preprocessing pipeline, the spatial footprints of all cells were manually verified to remove lens artifacts. Position data were inferred from this LED offline following recording using a custom written MATLAB (MathWorks) script and were manually corrected if needed. The experimenter manually segmented data recorded in the compartment and hallway, as well as the most recent entryway based on the recorded position data prior to all further analyses.

In one supplemental analysis, we characterized calcium transients by eschewing firing rate estimation via autoregressive deconvolution and instead binarized each trace as follows. dF/F traces were first Z-scored relative to their baseline variance, estimated from the half-normal distribution as when computing SNR. Next, all periods such that $Z > 3$ and $dZ/dt > 0$ were assigned a value of one. All other periods were assigned a value of zero. This binarization procedure thus captures the rise times of high-confidence calcium transients.

**Data analysis**. All analyses were conducted using the vector of inferred likelihood of spiking events (ILSE), treating this vector as if it were the firing rate of the cell[19], except for one supplemental analysis, as described above.

Rate maps of activity in the compartment were constructed by first binning the position data into pixels corresponding to a 2.5 cm × 2.5 cm grid of locations. To construct a rate map, the mean ILSE was computed for each pixel and then smoothed with a 4 cm standard deviation isometric Gaussian kernel. Rate maps of activity in the hallway were created by first collapsing the position data onto a line drawn through the center of the hallway. Next, the position data on this line were binned into 17 equally spaced pixels. Next, the mean ILSE was computed for each pixel. This map was then smoothed with a 2 pixel standard deviation Gaussian kernel.

To identify place cells coding for locations within the compartment in a manner to avoid a bias for or against the observation of remapping, we applied the following procedures. First, we partitioned the data within the compartment according to the most recent entryway, and according to the first and second halves of the recording and created rate maps for each partition. Next, we computed the Pearson correlation between each pair of cross-half rate map comparisons (e.g., Entry A half 1 to Entry A half 2; Entry A half 1 to Entry B half 2; Entry B half 1 to Entry A half 2; Entry B half 1 to Entry B half 2), while subsampling the data to match the spatial sampling distributions (described below). Next, for each cell, we created a null distribution of chance correlation values by circularly shifting the ILSE vector in time a random amount at least 60 s away from its proper alignment and recomputing all four correlations 500 times. The maximum correlation value of each iteration was then compared to the maximum value of the cell's actual correlations. A cell was categorized as a place cell if the maximum of its actual correlations exceeded the 95th percentile of this shuffled control. This selection procedure thus inherently corrects for multiple comparisons and weighs correlations within and across both most recent entryways equally. This procedure identified as place cells an average of 29.3% ± 1.8% (SEM) with split-half reliable compartment rate maps in the initial CA1 recordings, 39.2% ± 1.9% in the CA3 recordings, 27.7% ± 1.4% in the recordings of experimental mice with hm4di (26.9% ± 1.9% under Saline; 28.5% ± 2.0% under CNO), and 28.5% ± 1.8% in the recordings of control mice without hm4di (24.6% ± 2.3% under Saline; 32.1% ± 2.5% under CNO).

Signal-to-noise (SNR) for each dF/F trace was computed as a measure of data quality prior to spike estimation via deconvolution as described in previously[31]. Briefly, because calcium transients around the baseline can only be positive, for each trace the variance of a normal noise distribution was estimated by scaling the standard deviation of trace values below baseline via a half-normal distribution, such that:

$$\text{NOISE} = \frac{\text{std}(\mathbf{t}(\mathbf{t} < 0))}{\sqrt{1 - \frac{2}{\pi}}} \tag{1}$$

where $t$ is the detrended filtered calcium trace and $\mathbf{t}(\mathbf{t} < 0)$ are all filtered trace values below baseline. Next, we $z$-score the trace $\mathbf{t}$ such that $z(\mathbf{t}) = \mathbf{t}/\text{NOISE}$. These $z$-scored values are next transformed into probabilities $p(\mathbf{t}) = \varphi(-z(\mathbf{t}))$, where $\varphi$ denotes the cumulative distribution function of the standard normal distribution. Then we compute the lowest probability event over a time window of 0.4 s ($N = 12$ frames), the duration around which a GCamp6F transient is typically at its maximum, such that:

$$p_{\min} = \min_{i}\left(\prod_{j=0}^{N-1} p(\mathbf{t}(i+j))\right)^{1/N} \tag{2}$$

where $\min_i$ is the minimum value across all timepoints $i$ from $i = 1$ to $i =$ the length of $\mathbf{t} - N + 1$. Finally, we define the average peak SNR as:

$$\text{SNR} = -\varphi^{-1}(p_{min}) \tag{3}$$

where $\varphi^{-1}$ is the quantile function of the standard normal distribution (logit function).

Spatial information content (SIC) was computed from the whole-session rate maps of each cell as described previously[32] via the equation:

$$\text{SIC} = \sum_i \mathbf{s}_i \mathbf{r}_i \log\left(\frac{\mathbf{r}_i}{\bar{\mathbf{r}}}\right) \tag{4}$$

where $i$ is the rate map pixel index, $\mathbf{s}_i$ is the probability of sampling pixel $i$, $\mathbf{r}_i$ is the mean firing rate at pixel $i$, and $\bar{\mathbf{r}}$ is the mean firing rate across all pixels.

To correct for biases in sampled spatial locations, we subsampled our data during all comparisons to match the spatial sampling distributions across all comparisons (Supplementary Fig. 6). To do so, we computed the minimum number of samples recorded at each pixel location across all comparisons. Next for each comparison, we included a random subset of the data recorded at that location to match that minimum number of samples. Because this random subsampling introduces variability, we repeated all comparisons 100 times and took the mean value as the final measure of that comparison.

The primary measure of interest in most comparisons was population vector similarity assessed via a population vector correlation. This measure was computed by first concatenating the pixel values of all rate maps and all cells into a large vector for each comparison. Next, the Pearson correlation between these two vectors was taken. Similar results were observed for all analyses when cosine distance was instead used to assess population similarity (not shown). To quantify changes in the relative firing rates at an individual cell level, we computed the absolute percent change in firing rate within the compartment across both halves of the recording when entering through either the same entryway or different entryways. Mean firing rates were defined as the firing rate averaged over the entire time spent within the compartment following each entryway for each half independent of position (after downsampling to correct for differences in the spatial sampling distribution). Firing rates in each half were normalized to the maximum firing rate for that half. For example, the absolute percent change in firing rate between entering from entryway A in the first half and entryway B in the second half $\Delta F^{A,B}$ was computed as:

$$\Delta F^{A,B} = \left| \frac{F_1^A}{\max(F_1^A, F_1^B)} - \frac{F_2^B}{\max(F_2^A, F_2^B)} \right| \tag{5}$$

where $F_1^A$ denotes the firing rate when entering from entryway A in the first half of the recording, $F_2^B$ denotes the firing rate when entering from entryway B in the second half of the recording, etc.

**Histological validation of expression and recording targets**. Once mice had completed all behavioral experiments, animals were perfused to verify GRIN lens placement and virus expression. Mice were deeply anesthetized and intracardially perfused with 4% paraformaldehyde in PBS. Brains were dissected and post-fixed with the same fixative. Coronal sections (40 μm) of the entire hippocampus were cut using a vibratome and sections were mounted directly on glass slides. Sections were split and half of all sections were stained for cresyl violet to localize GRIN lens placement, the other half of sections were stained for DAPI and mounted with Fluromount-G (Southern Biotechnology) to evaluate virus expression.

To evaluate the hm4di-mCherry fluorescence for each mouse, pyramidal cells were quantified across eight coronal slices representing the injection sites along the septo-temporal axis of the hippocampus, with three randomly selected images taken across

three subregion (CA1, CA3, dentate gyrus) for a total of 72 images analyzed per animal. To evaluate the level of expression in each subregion, the number of transfected pyramidal cells was counted using imageJ (version 1.51j8) and virus expression was imaged with an AxioObserver.Z1 microscope (Carl Zeiss).

**Statistics and reproducibility**. All statistical tests are noted where the corresponding results are reported throughout the main text and supplement. All tests were uncorrected 2-tailed tests unless otherwise noted. $Z$-values for nonparametric Wilcoxon tests were not estimated or reported for comparisons with fewer than 15 datapoints. In all figures, the mean served as the measure of central tendency, and error bars reflected ±1 standard error of the mean, unless otherwise noted.

**Reporting summary**. Further information on research design is available in the Nature Research Reporting Summary linked to this article.

## Data availability

The complete dataset for all experiments are publicly available at https://doi.org/10.5061/dryad.crjdfn31g or via request to the corresponding authors. Source data for all main text and Supplementary Figures can be found in the accompanying Source data Excel spreadsheet as noted in the figure captions. Source data are provided with this paper.

## Code availability

All custom code written for reported analyses are publicly available at https://github.com/akeinath/EntrywayRemapping or via request to the corresponding authors.

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

## Acknowledgements

We kindly thank S. Kim and P. Mannarino for technical assistance and J. Hinman, A. Alexander, S. Chekhov, R. Rozeske, J. Ying, C. Kramar, and M. Hasselmo for comments of previous versions of this paper. We especially thank D. Aharoni for extensive guidance in using the UCLA miniscope. This work was supported by a McGill University Healthy Brains for Healthy Lives CFREF postdoctoral fellowship to A.T.K., Fonds de Recherche du Québec—Santé (FRQS) postdoctoral fellowship to A.N.P., and funding from the Canadian Institutes for Health Research grants (#367017 and #377074), the Natural Sciences and Engineering Council (Discovery grant #74105), and the Canada Research Chairs Program to M.P.B.

## Author contributions

A.T.K. contributed to experimental design, surgeries, recordings, analysis of data, as well as drafting and revising the paper. A.N.P. contributed to surgeries and recordings. J.C.R. contributed to histological verification of recording sites and quantifications. M.P.B. contributed to experimental design, analysis of data, as well as drafting and revising the paper.

## Competing interests

The authors declare no competing interests.
