## [Peer Review File · Nature Communications]

Reviewers' Comments:

Reviewer #1:

Remarks to the Author:

In this study, A.T. Keinath et al. examine the effect of latent information on population coding in the hippocampus, specifically in areas CA1 and CA3. This work follows studies with similar scope from the Eichenbaum and Hasselmo labs (e.g. Wood et al., 2000, *Neuron*; Griffin et al. 2007, *The Journal of Neuroscience*) that tried to assess the influence of latent factors on representations in the hippocampus. However, in this study the authors are trying to focus on a purely mnemonic influence on population activity in hippocampus by using a novel behavioral task in which animals explore an environment, which features an angled corridor in one corner through which animals can move freely in both directions (a task and goal free environment). The authors claim that the population code in CA1 and CA3 inside the environment (box) is influenced by the most recent history of where the animals exited the corridor in the absence of any other, sensory or task related changes. For this, they employ in vivo calcium imaging via head mounted one photon (epifluorescence) microscopes in animal cohorts implanted with GRIN lenses in either CA1 or CA3. They observe that on the population level, CA1 and CA3 activity is more dissimilar when compared across exit conditions than within. They then use a chemogenetic silencing approach in transgenic animals to elucidate the role of DG/CA3 activity in causing this effect. They find that silencing large areas of DG/CA3 circuitry in these animals leads to an apparent stabilization of population vectors and thereby decreases the effect of recent exit path on population activity in CA1. These results are intriguing and are challenging the view of the hippocampal-entorhinal system of being able to maintain unified representations (building a coherent cognitive map) within an environment the animal explores. The authors use an array of challenging in vivo experiments, combined with a novel behavioral task and clever analysis to go after this puzzle. However, I have some concerns about both the raw data and its interpretation that I want to address below.

Comments by the reviewer:

Although miniaturized one photon epifluorescence imaging is a technique that has been widely adapted by the community and therefore feels like a "plug and play" method, the quality of recordings nevertheless has to be assessed on an experiment to experiment basis. Objective thresholds for cell inclusion based for example on signal to noise ratio, baseline drift (either due to z movement or bleaching induced) are not described. It is also not described how stability of sessions was evaluated, specifically whether efforts were taken to rationalize cutoffs for motion correction quality (were any sessions discarded because they were too unstable?). The authors use an autoregressive model for spike deconvolution: Did the choice of parameters influence the results (place cell detection, etc.)? Moreover, the only "raw" data shown are filtered traces (Fig. 1 c). It would be desirable to show more of the raw recordings (both fluorescence traces as well as in vivo images with extracted cell maps) in conjunction with histology. It is really hard to assess the quality of the implants from histology shown in Figure 1b, which shows an outline of an implanted GRIN lens against an almost completely black background. Taken together it is very hard to judge how good the raw data really was.

The biggest concern here is that the inherent non-linearities of calcium imaging plus instability of some recordings, which in turn could lead to cells dropping in and out of existence, could lead to an overestimation of the observed effects that seems very small to begin with (Fig. 1e). From the figures presented it is unclear whether this effect might have been driven by only a few of animals or was equally variable in all of them.

Regarding the extraction of place cells itself, the authors employ a stability metric (pearson's correlation between adjacent session halves / conditions), but criteria like spatial information content, etc. that would distinguish these cells further from cells with stable ratemaps without any obvious spatial selectivity are not used. In line with this, for all population vector analyses, which include all

recorded cells regardless of functional tuning properties, it would be helpful to see whether the ratio of spatial information in a given session (on the population level) influences the result (e.g. place cells only vs. all cells).

Although the behavioral apparatus is quite simple, it is a novel paradigm not – to my knowledge – described by any other group. It would therefore be desirable to examine the animal's behavior more thoroughly. Specifically: How were exits from entryway A and B distributed throughout the session? It would help to see a timeline to assess what proportion of exits could be attributed to either entryway and how many of them occurred during the first half vs. the second half of the recording. The quality of the analysis would increase a lot knowing that both conditions were more or less equally represented and thereby did not suffer from behavioral biases.

It would also help to assess the normalization procedure explained in Line 89 and FigureS1 more thoroughly: "To control for these possible biases, we subsampled the data to match the sampling distributions across all conditions prior to all analyses". It is unclear how much data (coverage) actually remained over sessions and animals after this "minimally viable set" was extracted (only one example shown in Figure S1 a). Were any sessions excluded because they did not meet a minimum amount of coverage? In line with this: Could Figure 1d and Figure 2a (all ratemaps) be shown after this correction procedure? Since all subsequent analysis depends on it, it would be informative to present the ratemap data after this correction has been applied.

For analysis presented in Figure 3 (and described in manuscript Line 120 and following) I am not sure I understand what the authors did there. It seemed strange that the splitting on "two-back" trials does not replicate the results of splitting on entryway information, specifically since the remapping is already present in the corridor (as shown in Fig. S1 g/h): Since the animals do not turn around inside the corridor (as described by the authors, Line 439), isn't the outcome predetermined from when the animal first enter the corridor?

Regarding the experiments involving DREADD mediated silencing in Grik4-cre animals it seems strange that the original mouse line is advertised both by the Jackson laboratory as well as the paper by Nakazawa et al. (Science, 2002) as CA3 specific line (not DG and CA3 specific as described by the authors) and from the histology shown in Figure S3 it seems that there is a marked absence of expression of syn-DIO-hm4DI-mCherry in dorsal CA3. How do the authors explain this discrepancy, and, since dorsal CA1 was targeted for the imaging experiments, does that not enforce a reinterpretation of their data? As with the histology shown in Figure 1 (see comments above), the quality of the data presented in Figure S3 does not allow a careful assessment of where the focus of expression actually was in these animals.

In light of a recent study (Davoudi, Foster, Nature Neuroscience, 2019), it seems surprising that no detrimental effects on place cell stability in CA1 were found after acute CA3 silencing (Figure 4). The authors of this recent study acutely silenced CA3 output and measured strong effects on CA1 place cell stability, identifying CA3 as the predominant driver of CA1 place cell activity. It is also not clear why the authors focus on silencing CA3 in the first place in conjunction with DG, since they observe similar effects in CA1 in CA3. Following their logic, wouldn't it more plausible to suspect DG then as driver of these differences?

Also, it would be helpful to understand the authors' evaluation on why remapping inside the "linearized" corridor segment seems to be unaffected by CNO mediated silencing. How exactly do the authors suppose these phenomena (remapping on entry way vs. remapping due to reversed traversal of a linear track within the same environment) are different in their task?

For the general assessment of the study result it would be great to understand the authors' viewpoint on why they suspect that this latent information in the absence of task structure should be present in

the first place? What ethological relevance does it have for the animal? One major concern here is that the observed small differences and variability of the effect could be explained by, for example, (variable) time of total exposure to this environment. The authors do not describe extensive pretraining of animals in the environment, which means that this environment indeed seemed to be novel at first and animals got familiar with it over time. Novelty effects as well as behavioral changes afflicted to this might explain why in Figure S2 d (recordings in the larger environment) there is a trend towards more stable population representations in later sessions. If that is the case, the authors should exclude that stabilization of the cognitive map over time (with increasing familiarity) could ameliorate the observed differences.

Minor points:

- Figure 1b. Histology and cell map: no scale bars are shown
- Figure 1c. no scale bars for traces are shown
- Figure S1 a: no scale bars are shown
- Figure S1 b, d: Axis labels are not shown
- Figure S3 b. No scale bars shown for histology

Reviewer #2:

Remarks to the Author:

In this manuscript, Keinath et al. set out to test whether the activity of hippocampal place cells reflects recent behaviour or experiences – which they term latent information. To do this, they recorded the activity of ensembles of CA1 and CA3 place cells using wide-field fluorescent microscopy, as mice explored an arena which they accessed by two different entryways on different sides of the arena, connected by a hallway. Within a recording session, the mice were free to explore the whole environment, and so entered the arena multiple times from each entry way. By partitioning the data based on the most recent entryway, the authors could compare place cell activity in the arena when the animal entered via the two different entryways. The data convincingly show that place cells in both CA1 and CA3 fired at different rates within the arena dependent on which entryway the mouse used to enter the arena. This entryway-dependent “rate remapping” of the arena was evident for at least 5 s after the animals entered the arena, and also appeared to be distributed across the arena – i.e. did not only occur in cells that fired very close to the entryways.

This finding is important, as it shows that during free exploration of an environment, the activity of hippocampal place cells reflects previous experience. It builds on a previous study by Ferbinteanu and Shapiro (Neuron. 2003 Dec 18; 40(6):1227-39 – not referenced in the current manuscript) who showed rate remapping in place cells on a plus maze in rats. In this case, the activity on a given goal arm of the maze (e.g. west arm) was dependent on whether the rat had come from the north or the south start arm of the maze prior to entering the west arm (termed by them retrospective activity). Thus, the finding that the activity of place cells reflects previous experience is not novel. The current finding extends this by showing that it also occurs during free exploration of an environment, whereas in the Ferbinteanu task, the rats were in a goal directed task (although memory of the start arm was not required for accurate behaviour, so could equally be considered “latent” information).

What takes the current findings significantly beyond those of Ferbinteanu and Shapiro is that Keinath et al then go on to show that chemogenetic inhibition of DG and CA3 cells (targeted by using a Grik4-cre mouse line which expresses cre in DG and CA3 but not in CA1) eliminates the entryway – dependent remapping of CA1 place cells, while not affecting the spatial tuning of the cells. This

indicates that the DG-CA3 circuitry plays an essential role in driving the rate remapping of CA1 (and CA3) cells driven by previous experience (the entryway) in this study. This is important, as previous studies have found that direct inputs from nucleus reuniens to CA1 drive CA1 place cell remapping that reflects planned/future trajectories, so provides evidence that different pathways support hippocampal place cell remapping based on future versus past experiences. This has important implications for understanding the circuitry supporting spatial memory.

Overall the experiments are well designed and carefully executed, the data are clearly and completely described, appropriate control procedures have been conducted (such as sub-sampling data to match spatial sampling distributions for comparison of firing rate maps) and the findings are robust. I have several minor comments (below) but there are no major problems or issues.

1. The papers by Ferbinteanu & Shapiro (2003) and Frank et al (2000) demonstrating “retrospective” coding of hippocampal place cells should be cited and discussed – they show similar remapping based on previous experience, albeit in a goal-directed task.
2. Methods related to calculation of firing rates need to be more clearly explained. In the description of this (lines 518-527), it is not clear what measure of firing rate is used for each split half condition (F1A, F1B, F2A, F2B). Is it a simple mean firing rate (number of “spikes” in F1A/time in F1A)? Or is it the mean firing rate across the pixels of the rate map that is generated for that condition? Or perhaps the mean firing rate within the place field? Having calculated the mean firing rate, for each condition, it IS clear from the description how these are then normalised.
3. Figure 1e – state in the legend that each dot represents the population vector correlation for one session from one mouse. Data are 29 sessions from 5 mice.
4. Line 103-104 – make it clear why entry-specific firing is expected in the hallway (presumably because it is direction-selective).
5. Lines 121-123 – rationale is not quite clear for the two-back analysis – it would be helped if you made a specific (and well justified) prediction here before describing the data. For example, if the CA1 is encoding information about previous experiences, then not only the most recent entry but also the previous entry might also be influencing firing.
6. Lines 123-124 and figure 3b – it appears that locations closer to the entryways may have been more likely to show significant entry-way specific differences. There is no statistical test reported to determine whether the distribution of differences in firing rates were uniform across the arena, or were clustered in particular areas.
7. Figure 3c (and S3d) – a clearer explanation of how the temporal profile is generated. In the text it suggests that PVs-PVd is calculated when only including data from progressively longer times since entry. How is this actually done? Does this mean that the values of PVs-PVd at the 5s time point ONLY include data from at least 5s after the animal entered the arena, at the 4s time point is only data from at least 4s after entry etc? If this is the case, it supports the authors claim that rate remapping continues for at least 5s. However, if progressively later time points since entry are being added in (starting from short to long) then it is possible that the PVs-PVd difference shown for the longer time since entry could primarily be driven by earlier time points. I think this just requires clarification. Also, how long are the incremental time bins?
8. Figure 4b – the example firing rate maps are from different cells in the saline vs CNO conditions. It would be more convincing to also show the same cells under saline and CNO, to show changes at the

level of specific individual place cells.

9. Figure S1b and d – place field locations are used, but it is not clear how place fields are defined.

10. Discussion/introduction – throughout the manuscript the authors refer to the entry-way specific rate remapping as reflecting “latent” information. Does this just mean it reflects previous experience? And if not, how can the authors distinguish between “non-latent” and “latent” previous experience? Related to this, the authors do not discuss what aspects of the previous experience are likely to be driving the rate remapping. One simple explanation is that the direction of entry into the arena is driving the rate remapping (as drawn the entries are from the north and east of the arena). This would be consistent with previous data from Grieves et al (cited in the current manuscript) that place cells remap less between identical compartments whose entryways are facing in the same direction (e.g. all in the south wall), than the same compartments when the entryways are oriented differently. Some discussion of this possibility is warranted.

Reviewers' comments:

Reviewer #1 (Remarks to the Author):

In this study, A.T. Keinath et al. examine the effect of latent information on population coding in the hippocampus, specifically in areas CA1 and CA3. This work follows studies with similar scope from the Eichenbaum and Hasselmo labs (e.g. Wood et al., 2000, *Neuron*; Griffin et al. 2007, *The Journal of Neuroscience*) that tried to assess the influence of latent factors on representations in the hippocampus. However, in this study the authors are trying to focus on a purely mnemonic influence on population activity in hippocampus by using a novel behavioral task in which animals explore an environment, which features an angled corridor in one corner through which animals can move freely in both directions (a task and goal free environment). The authors claim that the population code in CA1 and CA3 inside the environment (box) is influenced by the most recent history of where the animals exited the corridor in the absence of any other, sensory or task related changes. For this, they employ in vivo calcium imaging via head mounted one photon (epifluorescence) microscopes in animal cohorts implanted with GRIN lenses in either CA1 or CA3. They observe that on the population level, CA1 and CA3 activity is more dissimilar when compared across exit conditions than within. They then use a chemogenetic silencing approach in transgenic animals to elucidate the role of DG/CA3 activity in causing this effect. They find that silencing large areas of DG/CA3 circuitry in these animals leads to an apparent stabilization of population vectors and thereby decreases the effect of recent exit path on population activity in CA1. These results are intriguing and are challenging the view of the hippocampal-entorhinal system of being able to maintain unified representations (building a coherent cognitive map) within an environment the animal explores. The authors use an array of challenging in vivo experiments, combined with a novel behavioral task and clever analysis to go after this puzzle. However, I have some concerns about both the raw data and its interpretation that I want to address below.

We thank the reviewer for a useful appraisal and for highlighting several important opportunities to strengthen this work.

Comments by the reviewer:

Although miniaturized one photon epifluorescence imaging is a technique that has been widely adapted by the community and therefore feels like a “plug and play” method, the quality of recordings nevertheless has to be assessed on an experiment to experiment basis. Objective thresholds for cell inclusion based for example on signal to noise ratio, baseline drift (either due to z movement or bleaching induced) are not described. It is also not described how stability of sessions was evaluated, specifically whether efforts were taken to rationalize cutoffs for motion correction quality (were any sessions discarded because they were too unstable?). The authors use an autoregressive model for spike deconvolution: Did the choice of parameters influence the results (place cell detection, etc.)? Moreover, the only “raw” data shown are filtered traces (Fig. 1 c). It would be desirable to show more of the raw recordings (both fluorescence traces as well as in vivo images with extracted cell maps) in conjunction with histology. It is really hard to assess the quality of the implants from histology shown in Figure 1b, which shows an outline of an implanted GRIN lens against an almost completely black background. Taken together it is very hard to judge how good the raw data really was. The biggest concern here is that the inherent non-linearities of calcium imaging plus instability of some recordings, which in turn could lead to

cells dropping in and out of existence, could lead to an overestimation of the observed effects that seems very small to begin with (Fig. 1e). From the figures presented it is unclear whether this effect might have been driven by only a few of animals or was equally variable in all of them.

We agree with the reviewer that it is especially important to check and include a variety of measures by which the quality of the raw calcium imaging data can be assessed. Prior to our submission we confirmed that our results were indeed robust to parameterizations and cell selection criteria which we did not include in our initial submission. We have now made a number of changes in our revision to include these results. Specifically,

We report median signal-to-noise (SNR) values for all recordings (Fig. S2). SNR values were generally high (mean ~8 to 12) and the SNRs of all cells well exceeded minimum thresholds suggested by calcium imaging data analysis pipelines using this measure (SNR>2; CalmAn; Giovannucci et al., 2019).

We report changes in SNR across session halves as an assay of bleaching / drift for all recordings (Fig. S2). SNR remained high in the second half of recording, though an attenuation in SNR of ~10 to 20% was observed. Notably, no differences in SNR or change in SNR were observed between Saline and CNO sessions, indicating that neither absolute recording quality nor changes in recording quality across a session can account for the elimination of entryway remapping during CNO trials in experimental mice expressing hm4di.

We report mean frame images, extracted spatial footprints, and raw calcium traces for random subsets of cells for an example session from each animal in each experiment (Fig. S1).

We report animal-wise results demonstrating the reliability of these effects within individual animals for our primary measures of interest (Figs. S7-8).

We report the percentage of cells meeting place cell criteria for each animal for all experiments (Fig. S3).

Report that entryway remapping results and the effects of DG-CA3 circuit inhibition hold when alternative methods are used to characterize calcium transients (Fig. S11). Specifically, we eschewed autoregressive deconvolution entirely and binarized traces based on their calcium dynamics, such that periods of high-and-increasing dF/F were assigned values of one, while all other periods were assigned values of zero.

Regarding the extraction of place cells itself, the authors employ a stability metric (pearson's correlation between adjacent session halves / conditions), but criteria like spatial information

content, etc. that would distinguish these cells further from cells with stable ratemaps without any obvious spatial selectivity are not used. In line with this, for all population vector analyses, which include all recorded cells regardless of functional tuning properties, it would be helpful to see whether the ratio of spatial information in a given session (on the population level) influences the result (e.g. place cells only vs. all cells).

To clarify, all analyses (including PV differences) in our initial submission included only the cells meeting our place cell selection criteria (split-half reliability exceeding chance). In our revision we now demonstrate that entryway remapping results and the effects of DG-CA3 circuit inhibition are robust to various cell selection criteria, including significant spatial information content and when all cells are included (Fig. S10).

Although the behavioral apparatus is quite simple, it is a novel paradigm not – to my knowledge – described by any other group. It would therefore be desirable to examine the animal’s behavior more thoroughly. Specifically: How were exits from entryway A and B distributed throughout the session? It would help to see a timeline to assess what proportion of exits could be attributed to either entryway and how many of them occurred during the first half vs. the second half of the recording. The quality of the analysis would increase a lot knowing that both conditions were more or less equally represented and thereby did not suffer from behavioral biases.

We now include a table characterizing the behaviors of each mouse including entryway bias, trajectory characteristics, and time and sampling coverage of the environment after downsampling to match spatial sampling distributions across entryways and halves, separated by condition where appropriate (Table S1). A significant bias in entryway preference was observed in 4 of 21 instances. Coverage after matching sampling distributions was generally high (60-85%). Importantly, none of these behavioral characteristics differed between Saline versus CNO sessions, indicating that such differences cannot account for the elimination of entryway remapping under CNO in experimental mice with hm4di.

It would also help to assess the normalization procedure explained in Line 89 and FigureS1 more thoroughly: “To control for these possible biases, we subsampled the data to match the sampling distributions across all conditions prior to all analyses”. It is unclear how much data (coverage) actually remained over sessions and animals after this “minimally viable set” was extracted (only one example shown in Figure S1 a). Were any sessions excluded because they did not meet a minimum amount of coverage? In line with this: Could Figure 1d and Figure 2a (all ratemaps) be shown after this correction procedure? Since all subsequent analysis depends on it, it would be informative to present the ratemap data after this correction has been applied.

We now include a figure showing heatmaps of mean spatial sampling after matching sampling distributions, as well as the same sampling distributions excluding data from the first 5 s of each trajectory (Fig. S5). In addition, we now include additional example rate maps from all recorded mice in the initial CA1 and CA3 recording experiments (Fig. S4), as well as example rate maps after one iteration of matching the behavioral sampling for these sessions, as suggested by the reviewer (Fig. S6). One CA1 mouse from the initial recording experiments was excluded prior to analysis for an extreme entryway bias (> 5:1) which persisted across all sessions; we now note this in the Methods.

For analysis presented in Figure 3 (and described in manuscript Line 120 and following) I am not sure I understand what the authors did there. It seemed strange that the splitting on “two-back” trials does not replicate the results of splitting on entryway information, specifically since the remapping is already present in the corridor (as shown in Fig. S1 g/h): Since the animals do not turn around inside the corridor (as described by the authors, Line 439), isn’t the outcome predetermined from when the animal first enter the corridor?

We appreciate the reviewer noting the need for clarification with this analysis. There was ambiguity in our wording and in our figure caption schematizing this analysis. The two-back analysis entailed splitting the data according to the second-most recent entrance into the compartment, not the prior entryway encountered when entering the corridor (which, as the reviewer noted, would be highly correlated with the most recent entryway). We have now modified the figure caption and the description of this analysis in the main text to clarify this ambiguity (lines 609 and 145).

Regarding the experiments involving DREADD mediated silencing in Grik4-cre animals it seems strange that the original mouse line is advertised both by the Jackson laboratory as well as the paper by Nakazawa et al. (Science, 2002) as CA3 specific line (not DG and CA3 specific as described by the authors) and from the histology shown in Figure S3 it seems that there is a marked absence of expression of syn-DIO-hm4DI-mCherry in dorsal CA3. How do the authors explain this discrepancy, and, since dorsal CA1 was targeted for the imaging experiments, does that not enforce a reinterpretation of their data? As with the histology shown in Figure 1 (see comments above), the quality of the data presented in Figure S3 does not allow a careful assessment of where the focus of expression actually was in these animals.

Indeed, we employed this mouse line and injection schedule with the hopes of specifically targeting CA3, expression in DG was unexpected. Given this unexpected expression, we crossed our Grik4 cre line with a constitutive DREADD line to ensure that this expression was in fact driven by the presence of cre within the DG. (The constitutive mouse line was ultimately abandoned due to the presence of hm4di receptors in other internal organs). Transfection did not exclude dorsal CA3, as is visible in examples such as in Fig. 4a, and below:

Furthermore, for a separate project another student quantified the transfection given our injection schedule and found that expression persisted throughout the AP axis of both CA3 and DG, as shown below:

Figure 1. Quantification of viral expression in hippocampal subregion CA3. Panel A, Examples of a series of coronal brain sections of both hemispheres (anterior to posterior) used to quantify DREADDs expression (red). Panel B, for each section, regions of interests corresponding to the various hippocampal layers as well as control regions were drawn and fluorescence levels automatically calculated. Panel C, heat maps displaying the varying level of fluorescence within each layer along the septo-temporal axis, for both the left and right hemispheres (so = Stratum Oriens; sp = Stratum Pyramidale; sr = Stratum Radiatum; lm = Stratum Lacunosum-Moleculare; lu = Stratum Lucidum; po = Polymorphic Layer; sg = Stratum Granulosum; mo = Stratum Moleculare). The unit of the fluorescence intensity is arbitrary and is a relative measure.

In light of a recent study (Davoudi, Foster, Nature Neuroscience, 2019), it seems surprising that no detrimental effects on place cell stability in CA1 were found after acute CA3 silencing (Figure 4). The authors of this recent study acutely silenced CA3 output and measured strong effects on CA1 place cell stability, identifying CA3 as the predominant driver of CA1 place cell activity.

Although our results are at odds with the Foster's result, our results are consistent with prior lesion studies such as (Brun, Moser 2002) in which no deficit in place coding or place recognition was observed following transection of CA1 inputs from CA3, as we note in our manuscript. It is possible that this discrepancy arises from the difference in the timescale of manipulations. Davoudi inhibited CA3 via optogenetics on short timescales, while both the Brun lesions and our DREADD manipulations persisted on significantly longer timescales.

It is also not clear why the authors focus on silencing CA3 in the first place in conjunction with DG, since they observe similar effects in CA1 in CA3. Following their logic, wouldn't it more plausible to suspect DG then as driver of these differences?

As we note above, expression in DG was unexpected, and as a result we are unable to adjudicate between more precise subcircuit-level hypotheses, such as whether CA3, the DG, or both specifically mediate the entryway remapping we observe. As the reviewer notes, we consider it plausible that the DG might be the primary driver of entryway remapping, given prior work implicating the DG in pattern separation more generally. Conversely, we also consider it plausible that CA3 might be necessary for maintaining latent information, given prior work implicating the dense recurrent circuitry within this subregion as contributing to mnemonic encoding. Finally, it's possible that both regions together make a unique but necessary contribution. Moreover, we note that the trisynaptic circuitry suggests that elimination of CA3 would also entail a loss of information from the DG to CA1, further complicating interpretation of a specific CA3 inhibition experiment. Ultimately, we agree with the reviewer that these are interesting and important hypotheses to be tested, but believe that they are beyond the scope of our current study.

Also, it would be helpful to understand the authors' evaluation on why remapping inside the "linearized" corridor segment seems to be unaffected by CNO mediated silencing. How exactly do the authors suppose these phenomena (remapping on entry way vs. remapping due to reversed traversal of a linear track within the same environment) are different in their task?

Although there are multiple differences between navigation within the compartment and navigation in the hallway, we believe that the primary driving difference may be the following. In the hallway, movement is constrained such that external cues are consistently experienced in a set pattern; thus within the hallway we suspect that remapping is driven by immediately-available external cues, as is typically seen on linear tracks (e.g. Navratilova, McNaughton, 2012). Once within the compartment, however, behavior is unconstrained and immediate external cues therefore do not disambiguate the most recent entryway. We now highlight this result and its possible interpretation explicitly in our discussion.

For the general assessment of the study result it would be great to understand the authors' viewpoint on why they suspect that this latent information in the absence of task structure should be present in the first place? What ethological relevance does it have for the animal? One major concern here is that the observed small differences and variability of the effect could be explained by, for example, (variable) time of total exposure to this environment. The authors do not describe extensive pretraining of animals in the environment, which means that this environment indeed seemed to be novel at first and animals got familiar with it over time. Novelty effects as well as behavioral changes afflicted to this might explain why in Figure S2 d (recordings in the larger environment) there is a trend towards more stable population representations in later sessions. If that is the case, the authors should exclude that stabilization of the cognitive map over time (with increasing familiarity) could ameliorate the observed differences.

We agree with the reviewer that these questions warrant additional consideration in our manuscript. We

have now added a paragraph in the discussion to address these and related questions (beginning at line 225). In brief, we suggest that at a neural level such representation may be driven by the unique state-space induced by the stereotyped navigation through the hallway, and that entryway remapping may follow from reinforcement learning models. We also raise the possibility that representation of latent variable during unconstrained navigation may be crucial for discovering contingencies beyond current spatial location itself. We further highlight the possibility that repeated experience may serve to stabilize the compartment map, though we note that in our data entryway remapping persevered for up to 21 sessions of recording and following repeated exposure to a non-remapping map in the case of mice with hm4di under CNO. We also note that, although the observed effect size is small, our results are consistently observed within individual animals and are robust to many features of the analysis, such as cell selection and the particular quantification of calcium transients (as noted above and included in the revised and additional supplementary figures).

Minor points:

- **Figure 1b. Histology and cell map: no scale bars are shown** - Added
- **Figure 1c. no scale bars for traces are shown** - Added
- **Figure S1 a: no scale bars are shown** - Added
- **Figure S1 b, d: Axis labels are not shown** - Added
- **Figure S3 b. No scale bars shown for histology** - Added

Reviewer #2 (Remarks to the Author):

In this manuscript, Keinath et al. set out to test whether the activity of hippocampal place cells reflects recent behaviour or experiences – which they term latent information. To do this, they recorded the activity of ensembles of CA1 and CA3 place cells using wide-field fluorescent microscopy, as mice explored an arena which they accessed by two different entryways on different sides of the arena, connected by a hallway. Within a recording session, the mice were free to explore the whole environment, and so entered the arena multiple times from each entry way. By partitioning the data based on the most recent entryway, the authors could compare place cell activity in the arena when the animal entered via the two different entryways. The data convincingly show that place cells in both CA1 and CA3 fired at different rates within the arena dependent on which entryway the mouse used to enter the arena. This entryway-dependent “rate remapping” of the arena was evident for at least 5 s after the animals entered the arena, and also appeared to be distributed across the arena – i.e. did not only occur in cells that fired very close to the entryways.

This finding is important, as it shows that during free exploration of an environment, the activity of hippocampal place cells reflects previous experience. It builds on a previous study by Ferbinteanu and Shapiro (Neuron. 2003 Dec 18; 40(6):1227-39 – not referenced in the current manuscript) who showed rate remapping in place cells on a plus maze in rats. In this case, the activity on a given goal arm of the maze (e.g. west arm) was dependent on whether the rat had come from the north or the south start arm of the maze prior to entering the west arm (termed by

them retrospective activity). Thus, the finding that the activity of place cells reflects previous experience is not novel. The current finding extends this by showing that it also occurs during free exploration of an environment, whereas in the Ferbinteanu task, the rats were in a goal directed task (although memory of the start arm was not required for accurate behaviour, so could equally be considered “latent” information).

What takes the current findings significantly beyond those of Ferbinteanu and Shapiro is that Keinath et al then go on to show that chemogenetic inhibition of DG and CA3 cells (targeted by using a Grik4-cre mouse line which expresses cre in DG and CA3 but not in CA1) eliminates the entryway –dependent remapping of CA1 place cells, while not affecting the spatial tuning of the cells. This indicates that the DG-CA3 circuitry plays an essential role in driving the rate remapping of CA1 (and CA3) cells driven by previous experience (the entryway) in this study. This is important, as previous studies have found that direct inputs from nucleus reuniens to CA1 drive CA1 place cell remapping that reflects planned/future trajectories, so provides evidence that different pathways support hippocampal place cell remapping based on future versus past experiences. This has important implications for understanding the circuitry supporting spatial memory.

Overall the experiments are well designed and carefully executed, the data are clearly and completely described, appropriate control procedures have been conducted (such as sub-sampling data to match spatial sampling distributions for comparison of firing rate maps) and the findings are robust. I have several minor comments (below) but there are no major problems or issues.

We thank the reviewer for a useful appraisal and for highlighting a number of important opportunities to strengthen this work.

1. The papers by Ferbinteanu & Shapiro (2003) and Frank et al (2000) demonstrating “retrospective” coding of hippocampal place cells should be cited and discussed – they show similar remapping based on previous experience, albeit in a goal-directed task.

The absence of these very relevant studies from our initial submission was an (embarrassing) oversight. We have now included these studies, and further elaborate on the relationship between these studies and our work in the discussion.

2. Methods related to calculation of firing rates need to be more clearly explained. In the description of this (lines 518-527), it is not clear what measure of firing rate is used for each split half condition (F1A, F1B, F2A, F2B). Is it a simple mean firing rate (number of “spikes” in F1A/time in F1A)? Or is it the mean firing rate across the pixels of the rate map that is generated for that condition? Or perhaps the mean firing rate within the place field? Having calculated the mean firing rate, for each condition, it IS clear from the description how these are then normalised.

We computed MFR from the simple mean firing rate. We’ve now included the following clarifying statement in the Methods Section: Mean firing rates were defined as the firing rate averaged over the

entire time spent within the compartment following each entryway for each half independent of position (after downsampling to correct for differences in the spatial sampling distribution).

3. Figure 1e – state in the legend that each dot represents the population vector correlation for one session from one mouse. Data are 29 sessions from 5 mice.

We have added this information to the figure captions for both Fig. 1e and Fig. 2b.

4. Line 103-104 – make it clear why entry-specific firing is expected in the hallway (presumably because it is direction-selective).

We have now explicitly stated that this expectation is prompted by the typical observation of direction-selective firing in linear track environments.

5. Lines 121-123 – rationale is not quite clear for the two-back analysis – it would be helped if you made a specific (and well justified) prediction here before describing the data. For example, if the CA1 is encoding information about previous experiences, then not only the most recent entry but also the previous entry might also be influencing firing.

We have now devoted a separate paragraph to these results to better motivate this analysis, and to elaborate on an interpretation of the results when considered in conjunction with prior work (beginning at line 141).

6. Lines 123-124 and figure 3b – it appears that locations closer to the entryways may have been more likely to show significant entry-way specific differences. There is no statistical test reported to determine whether the distribution of differences in firing rates were uniform across the arena, or were clustered in particular areas.

We now report that the magnitude of remapping was negatively correlated with the distance to the nearest entryway in CA1 ($r = -0.279$, $p = 4.920e-4$) but not CA3 ($r = 0.009$, $p = 0.917$) in our data.

7. Figure 3c (and S3d) – a clearer explanation of how the temporal profile is generated. In the text it suggests that PVs-PVd is calculated when only including data from progressively longer times since entry. How is this actually done? Does this mean that the values of PVs-PVd at the 5s time point ONLY include data from at least 5s after the animal entered the arena, at the 4s time point is only data from at least 4s after entry etc? If this is the case, it supports the authors claim that rate remapping continues for at least 5s. However, if progressively later time points since entry are being added in (starting from short to long) then it is possible that the PVs-PVd difference shown for the longer time since entry could primarily be driven by earlier time points. I think this just requires clarification. Also, how long are the incremental time bins?

The analysis we performed here was the former: The mean values at the 5 s time point only include data from at least 5 s after the animal entered the compartment, the values at 4 s only include data from at least 4 s since entering the compartment, etc. We have now changed the main text and figure caption to clarify this, and include an example statement in the figure caption. Incremental time bins were computed at every 0.25 s, which we note in the figure caption.

8. Figure 4b – the example firing rate maps are from different cells in the saline vs CNO conditions. It would be more convincing to also show the same cells under saline and CNO, to show changes at the level of specific individual place cells.

Because we compute all analyses within-session, we aimed to avoid across-session cell registration (which we could not always apply to these data with high certainty due to dense clustering of cells and, in some cases, changes in the field of view across days) for generating examples and instead used an unbiased method for generating examples from each session individually. We now include across-session examples from when the registration could be applied with high certainty (Fig. S15).

9. Figure S1b and d – place field locations are used, but it is not clear how place fields are defined.

We have now included the clarifying statement in the figure caption: Place field locations were computed as the center of mass of activity across the entire compartment rate map, and we have modified the axes of these figures to indicate that they reflect the center of mass in pixels.

10. Discussion/introduction – throughout the manuscript the authors refer to the entry-way specific rate remapping as reflecting “latent” information. Does this just mean it reflects previous experience? And if not, how can the authors distinguish between “non-latent” and “latent” previous experience? Related to this, the authors do not discuss what aspects of the previous experience are likely to be driving the rate remapping. One simple explanation is that the direction of entry into the arena is driving the rate remapping (as drawn the entries are from the north and east of the arena). This would be consistent with previous data from Grieves et al (cited in the current manuscript) that place cells remap less between identical compartments whose entryways are facing in the same direction (e.g. all in the south wall), than the same compartments when the entryways are oriented differently. Some discussion of this possibility is warranted.

We used the term latent information to describe previous experience which could not be attributed to immediate sensory cues or goal-oriented planning. We did not intend to make a distinction between latent and ‘non-latent’ previous experience. We agree with the reviewer that it is interesting to speculate about plausible specific drivers of entryway remapping, including the possible influence of entry direction (indeed, we suspected entry direction would be a strong driver based on the Grieves result, and it is part of the reason why we designed the maze as we had). We now devote a paragraph in the discussion to highlight these and related possibilities (beginning at line 225).

Reviewers' Comments:

Reviewer #1:

Remarks to the Author:

The authors have put substantial work in improving the manuscript and added supplementary figures in response to many of the points raised in the first round of reviews. However, the added information raised some additional questions that I want to address below.

The biggest point of concern is still the effect size. The authors included a new figure (supplementary figure 7) showing the data for individual mice. It seems that the described effect is not robust over sessions even for single animals.

Is there any explanation to this that can be derived from behavioral biases on these days (referring to Supplementary table 1) or SNR differences?

Considering the behavioral quantification (added in Supplementary table 1): Ratemaps were calculated over just 60-70 seconds of data on coverage as low as 60% (on average) – is this correct? This seems very low to begin with and might explain why the effect was so difficult to detect in the first place.

In their response, the authors included a new figure showing the expression pattern in their transgenics (Grik4-cre mice). I do not agree that there is expression visible in CA3 cells. In fact, pyramidal cells in CA3 seem to be spared (the intense fluorescence seems to originate from mossy fibers streaming into CA3 from DG). Could the authors confirm that this is the case? If true, this point should be made clear throughout results and discussion (e.g. lines 174/175, introduction lines 68,69) and statements involving the direct involvement of both DG and CA3 have to be formulated less strongly.

The histology showing implantation sides is still not shown. Although there seems to be no obvious correlation between effect size and putative amount of damage inflicted by the implantation, it is important to keep in mind that GRIN lenses with either 1.8 or 0.5 mm diameter, were used which are expected to inflict very different amounts of (cortical) damage during implantations. Although implantation coordinates are given, the position of where implants ended up might explain differences in effect sizes between animals (e.g. depending on where exactly CA1 was targeted for example).

The authors claim that the differences are robust to cell selection criteria (Supplementary figures 10 and 11). However, these criteria do mostly differentiate between putative place cells and non-place cells (either evaluated by significant split half correlations or significant spatial information content), and the observed effects hold true only when putative place cells are included (and not if all cells are considered). Maybe the authors could formulate this result differently to better reflect this result (e.g. lines 133,134,196).

Methods: Can the SNR calculation be described more clearly? How was baseline estimated? What does the 0.4 s(econds) window refer to?

Minor points:

It is stated that one CA1 animal was excluded from analysis and authors claim that this is in the methods section, but I cannot find this information.

Line 269. Typo: Nucleus Reuniens

Reviewer #2:

Remarks to the Author:

The authors have satisfactorily addressed all of the points I raised in the revised manuscript.

Reviewers' comments:

Reviewer #1 (Remarks to the Author):

The authors have put substantial work in improving the manuscript and added supplementary figures in response to many of the points raised in the first round of reviews. However, the added information raised some additional questions that I want to address below.

The biggest point of concern is still the effect size. The authors included a new figure (supplementary figure 7) showing the data for individual mice. It seems that the described effect is not robust over sessions even for single animals.

Is there any explanation to this that can be derived from behavioral biases on these days (referring to Supplementary table 1) or SNR differences?

Considering the behavioral quantification (added in Supplementary table 1): Ratemaps were calculated over just 60-70 seconds of data on coverage as low as 60% (on average) – is this correct? This seems very low to begin with and might explain why the effect was so difficult to detect in the first place.

It is indeed correct that following downsampling to match the sampling distributions sampling values are generally in that range. We would note that such coverage seems low due to this downsampling – assuming independent sampling of the compartment per entryway and half, 88% coverage for each session and half would yield 60% coverage following downsampling. Indeed, we agree that these limitations, also in part a product of limiting our recording times to 20 min to avoid serious adverse effects of bleaching the calcium signal, are less than ideal and may affect the power to detect this effect. Nevertheless, we think that downsampling in this way is crucial to ensuring that any results are not driven by differences in the spatial sampling distribution correlated with entryway, and we are confident that we have sufficient power to characterize entryway remapping given our sample sizes, for the reasons described below. As per the reviewer's suggestions:

We now include a figure (Supplementary Fig 14) showing the correlation between our main effect size and time in compartment, number of entries, duration per entry, entryway bias, cumulative distance per entry, data post sampling matching, area sampling post sampling matching, raw SNR, and the change in SNR over the two halves of recording separately for each dataset and condition. None of these measures consistently correlated with effect size. To further strengthen this claim, we also collapsed across all data, except for the mice with hm4di under CNO data, and observed no relationship between any of these variables and effect size:

These results indicate that variability in effect size is not driven by behavioral or SNR covariates.

To address concerns of robustness, we now include a bootstrapping analysis where we compute the proportion of significant comparisons we would have observed when drawing 10,000 random subsets (without replacement) from

our data across a range of 4 - 12 sessions (Supplementary Fig 14). This analysis revealed that all effects we report can be consistently observed (significant comparisons occurring for >99% of random subsets) for all conditions (except the mice with hm4di under CNO) with as few as 11 sessions included, and with even fewer in some conditions. This indicates that, though the effect size is generally small, the effect is robust and we have more than adequate power to observe it in our complete datasets.

We now include as a supplemental figure all main comparisons without downsampling to match the spatial sampling distribution (Supplementary Fig 15). We note that all observed effects of entryway remapping and the abolition of entryway remapping in mice with hm4di under CNO persist without the matching of sampling distributions. Indeed, some effects, such as the mean firing rate differences per cell by entryway, appear stronger. This gives us more confidence that the effects we observe are not a specific product of our downsampling to match spatial sampling distributions and may appear stronger without limiting the amount of data included.

In their response, the authors included a new figure showing the expression pattern in their transgenics (Grik4-cre mice). I do not agree that there is expression visible in CA3 cells. In fact, pyramidal cells in CA3 seem to be spared (the intense fluorescence seems to originate from mossy fibers streaming into CA3 from DG). Could the authors confirm that this is the case? If true, this point should be made clear throughout results and discussion (e.g. lines 174/175, introduction lines 68,69) and statements involving the direct involvement of both DG and CA3 have to be formulated less strongly.

With this particular virus, we have observed that the level of fluorescence in the terminal is much higher than the level of fluorescence at the soma. This has made it difficult to highlight somatic expression in whole hippocampus without oversaturating the images. For this reason, in the paper we quantified the number of cell bodies across the DG, CA3 and CA1 rather than overall fluorescence intensity. We understand that the images provided in the paper do not clearly show CA3 somatic expression, so we have therefore provided additional images which support the quantification shown in Supplemental Figure 18 that show individual somas at the level of the pyramidal cell layer. Included here are examples from each of the animals included in the dataset with CA3 and DG somas indicated with arrows. Indeed, the mossy fiber fluorescence is very high in these images, however there is still a substantial amount of CA3 soma expression in the pyramidal cell layer. We would also point to the enlarged images shown in figure 4 and Supplemental Figure 18 which show CA3 soma expression. It should be noted that with the images included below, we changed the level of exposure between enlarged images in order to increase the visibility of CA3 cells in the pyramidal cell layer. We also point out that fluorescence is observed throughout stratum radiatum of CA1, which reflects expression of the CA3 inputs to region CA1.

The histology showing implantation sides is still not shown. Although there seems to be no obvious correlation between effect size and putative amount of damage inflicted by the implantation, it is important to keep in mind that GRIN lenses with either 1.8 or 0.5 mm diameter, were used which are expected to inflict very different amounts of (cortical) damage during implantations. Although implantation coordinates are given, the position of where implants ended up might explain differences in effect sizes between animals (e.g. depending on where exactly CA1 was targeted for example).

We now include a supplementary figure with additional histological examples for each lens size and target location, as well as a figure summarizing the lens locations of our mice (Supplementary Figure 1). In general, there was little variability in 1.8 mm diameter GRIN locations, and more variability in 0.5 mm diameter relay lenses targeting CA3.

The authors claim that the differences are robust to cell selection criteria (Supplementary figures 10 and 11). However, these criteria do mostly differentiate between putative place cells and non-place cells (either evaluated by significant split half correlations or significant spatial information content), and the observed effects hold true only when putative place cells are included (and not if all cells are considered). Maybe the authors could formulate this result differently to better reflect this result (e.g. lines 133,134,196).

We agree with the reviewer that these additional criteria were implemented to further separate place cells from other cell types. We have changed the wording of these sections to reflect this. For completeness, we now include in a supplementary figure (12) a direct comparison between cells characterized as place cells and cells which do not meet criteria. We now note explicitly that all entryway remapping effects are more pronounced in the place cell population than the non-place cell population.

Methods: Can the SNR calculation be described more clearly? How was baseline estimated? What does the 0.4 s(seconds) window refer to?

We apologize for the lack of clarity of our initial SNR description. We have changed this section to more precisely state our quantification. The 0.4 second window is the window over which the likelihood of a particular event given the noise distribution is assessed as established in prior work (Giovannucci, Pnevmatikakis, eLife, 2019) and is derived to capture the duration over which a GCamp6f transient is typically at its maximum values.

Minor points:

It is stated that one CA1 animal was excluded from analysis and authors claim that this is in the methods section, but I cannot find this information.

This exclusion is stated in the *Subjects* section of the methods as: "One mouse was excluded from the initial CA1 recording experiments prior to analysis due to an extreme entryway bias (> 5:1) which persisted across all sessions." We now note this in the main text as well to highlight this exclusion: "(Figure 1b; n = 6 mice, *one mouse excluded for persistent entryway behavioral bias*, n = 5 analyzed, 29 sessions; Supplementary Table 1)"

Line 269. Typo: Nucleus Reuniens

Corrected.

Reviewer #2 (Remarks to the Author):

The authors have satisfactorily addressed all of the points I raised in the revised manuscript.

Reviewers' Comments:

Reviewer #1:

Remarks to the Author:

The authors have successfully addressed the questions / comments I had in my review.

Minor correction:

Supplementary Figure 1.: ...and 0.5 mm diameter relay lens placement in CA1 (bottom).

Should be CA3?